



# Physically-based model for gully simulation: application to the Brazilian Semiarid Region

Pedro Henrique Lima Alencar[1,2], José Carlos de Araújo[2], and Adunias dos Santos Teixeira[2]

[1]TU Berlin, Institut für Ökologie, 10587 Berlin, Germany
[2]Federal University of Ceará, Departamento de Engenharia Agrícola, Fortaleza, Brazil

**Correspondence:** Pedro Alencar (pedro.alencar@campus.tu-berlin.de); José Carlos de Araújo (jcaraujo@ufc.br)

**Abstract.** Gullies are one of the most relevant erosion processes, connected to land degradation and desertification, in special in arid and semiarid regions. Despite its role, gully erosion is neglected by many models and researches. This study presents a physically-based model for small permanent gullies, typical in the Brazilian Semiarid Region. The model consists of coupling two previous models, those by Foster and Lane (1983) and Sidorchuk (1999). As both models require input data of peak

discharge and duration, different rain intensities were tested. The rain intensity that suited gully erosion modelling best was the 30-minute intensity. The Foster and Lane model supplied a better response for smaller areas, where bed-channel erosion is more pronounced. The Sidorchuk model presented a better performance in larger sections, where wall erosion becomes more prominent. The experimental area is located in the semiarid State of Ceará, Brazil, in which the land use is characterised by agriculture and livestock. We measured and modelled three gullies ageing almost six decades. The threshold between the

prevailing domains of each process (channel bed or wall erosion) is based on the cross-section area; and it is intrinsically connected to wall erosion: for the case study, the threshold area was approximately 2 m$^2$. The final model, hereby called FL-SM (Foster & Lane and Sidorchuk Model) performed very well, with Nash-Sutcliffe coefficient of 0.846.

## 1   Introduction

On our way to sustainable development and environmental conservation, soil erosion by water was pointed out as a key problem to be faced in the 21$^{st}$ century (Borrelli et al., 2017; Poesen, 2018). The impact of water-driven soil erosion, on economy and food supply alone, represents an annual loss of US\$ 8 to 40 billion; a reduction in food production of 33.7 million tonnes; an increase in water consumption by 48 km$^3$. These effects are felt more severely in countries like Brazil, China and India; and in low-income households worldwide (Pimentel et al., 1995; Nkonya et al., 2016; Sartori et al., 2019). Estimations on total

investments to mitigate land-degradation effects on site (e.g. productivity losses) and their off-site effects (e.g. biodiversity losses, water body siltation) lead to more alarming values, averaging US\$ 400 billion yr$^{-1}$ (Pimentel et al., 1995; Nkonya et al., 2016). Nonetheless, those values were obtained by estimations of soil erosion using USLE (Universal Soil Loss Equation) or





similar methods, none considering gully erosion, thus the real economical and social impacts of soil erosion are not completely comprehended.

Notwithstanding, soil degradation had already been a national issue in the first years of the 20th century in the USA, for instance, being reported by the USDA and the National Conservation Congress, with over 44 thousand km² of abandoned land due to intense erosion. By the end of the 1930's this number had increased to over two hundred thousand km² (Montgomery, 2007). Among soil erosion mechanisms, gully erosion plays a relevant role in sedimentological processes in watersheds, since it frequently is the major source of sediment displacement (Vanmaercke et al., 2016). Ireland et al. (1939) observed early the

effect of intense land-use change on gully formation, mainly due to alteration on land-cover and flow path direction. These landscape modifications were connected to runoff acceleration and/or concentration, therefore, triggering gullies.

Gully erosion consists of a process that erodes one (or a system of) channel(s) that starts due to the concentration of surface water discharge erosion during intense rainfall events (Bernard et al., 2010). The concentrated flow causes a deep topsoil incision and may reach the groundwater table (Starkel, 2011). Gullies are connected to anthropogenic landscape modifications

and to land use and land cover changes, as observed in the Cerrado biome in Brazil (Hunke et al., 2015). On the other hand, the presence of vegetation may prevent soil erodibility both by increasing cohesion forces and enhancing soil structure (Li et al., 2017; Vannoppen et al., 2017). Maetens et al. (2012) suggested that land-use changes lead to runoff changes and, hence, directly affect erosive processes. Gully erosion can also be affected by climate change, e.g., an increase of rainfall intensity could lead to higher erosive potential (Nearing et al., 2004; Montenegro and Ragab, 2012; Figueiredo et al., 2016; Panagos

et al., 2017).

Gullies also play a relevant role in the connectivity of catchments (Verstraeten et al., 2006; Molina et al., 2009), allowing more sediment to reach water bodies and, thus, increasing siltation (de Araújo et al., 2006). Gullies are also strongly dependent of landscape factors. With the advance of machine-learning techniques and the use of large data sets, some of the factors that mostly influence gully formation were identified, such as lithology, land use and slope. Some indexes were also pointed as

relevant to indicate gully initiation, as the Normalized Difference Vegetation Index, Topography Wetness Index and Stream Power Index (Arabameri et al., 2018, 2019; Azareh et al., 2019).

For being particularly relevant among the erosion processes, gullies execute a great pressure on landscape development: they change the water-table height, alter sediment dynamics and increase runoff (Valentin et al., 2005; Poesen, 2018; Yibeltal et al., 2019). They represent an increasing risk to society and environment for affecting land productivity, water supply, floods,

debris flow and landslides (Liu et al., 2012, 2016; Ruljigaljig et al., 2017; Wei et al., 2018). Gullies also have a large impact on economy due to high mitigation costs, a reduction of arable fields, a decrease of groundwater storage, an increase of water and sediment connectivity and more intense reservoir siltation (Verstraeten et al., 2006; Pinheiro et al., 2016). The assessment of gully impacts on production costs in an arid region of Israel showed that costs of gully mitigation represent over 5% of total investments, and production losses are as large as 37 % (Valentin et al., 2005).

Despite their relevance to hydro-sedimentological processes, gullies are often neglected in models (De Vente et al., 2013; Poesen, 2018), and should be directly addressed (Paton et al., 2019). However, gully erosion is a process with the interaction of many variables, many of them difficult to assess (Bernard et al., 2010; Castillo and Gómez, 2016): according to Bennett



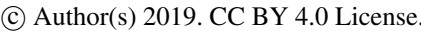


and Wells (2019), for instance, no model has ever been presented to clearly explain the process of gully formation. Among

the models that do consider gully erosion, the use of empirical approach prevails (Thompson, 1964; Watson and Laflen, 1986;

Woodward, 1999; Nachtergaele et al., 2001, 2002; Poesen et al., 2002; Yao et al., 2008; Wells et al., 2013); whereas others

focus primarily on physically-based algorithms (Foster and Lane, 1983; Storm et al., 1990; Hairsine and Rose, 1992; Ascough

et al., 1997; Sidorchuk, 1999; Alonso et al., 2002; Sander et al., 2007; Dabney et al., 2015).

It is, therefore, an important milestone to understand how gully erosion starts and develops (Poesen, 2018). The objective of

this work is to propose a physically-based model that predicts growing dynamics and sediment production in small permanent

gullies in a hillslope scale. Most of the basins around the world do not have subdaily rainfall data for obtaining the discharge

characteristics for storm events that cause gully erosion (de Araújo, 2007). As registered in literature by many authors (Ireland

et al., 1939; Katz et al., 2014), road construction is a main driver for gully initiation. It was assumed that small permanent

gullies are the result of active erosive processes that form channels by concentrated flow and do not interact with groundwater.

Normally, these gullies could be remediated by regular tillage process, but in abandoned or unclaimed land they usually remain

untreated for long periods.

## 2 MATERIALS AND METHODS

### 2.1 Study area

The Brazilian Semiarid Region (1 million $km^2$) is covered mainly by the Caatinga biome with predominantly bushes and

broadleaf deciduous vegetation. (de Araújo and Piedra, 2009; Pinheiro et al., 2013). The region is prone to droughts and highly

vulnerable to water scarcity (Coelho et al., 2017). More than 25 million people live in this region, where agriculture (maize,

beans, cotton) and livestock are of utmost socio-economic relevance. Usually, rural communities use deleterious practices, such

as harrowing and field burning, which enhance the risk of intense erosive processes. These characteristics lead to a scenario

of soil erosion and water scarcity with high social, economic and environmental consequences (Sena et al., 2014). Erosion in

general (and gullies in particular) increases local hydric vulnerability due to reservoir siltation (de Araújo et al., 2006) and

water-quality depletion (Coelho et al., 2017).

The study area is located in the Madalena Representative Basin (MRB, 75 $km^2$, state of Ceará, north-eastern Brazil; see

Figure 1), inserted in the Caatinga biome, a dry environment with a semiarid hot BSh climate, according to the Köppen

classification (Gaiser et al., 2003). The annual precipitation averages 600 mm, concentrated between January and June (Figure

2); and the potential evapotranspiration totals 2,500 $mm.yr^{-1}$. Geologically, the basin is located on top of a crystalline bedrock

with shallow soils and a limited water storage capacity. The rivers are intermittent and runoff is low, typically ranging from

40 to 60 $mm.yr^{-1}$. The basin is located within a land reform settlement with 20 inhabitants per $km^2$, whose main economic

activities are agriculture (specially Zea mays), livestock and fishing (Coelho et al., 2017; Zhang et al., 2018).

Three gullies were selected for this study, all located on the eastern portion of the basin. The studied gullies have the

following dimensions (average $\pm$ standard deviation): projection area (317±165 $m^2$), length (38 $\pm$ 6 m), volume (42 $\pm$ 25

90 $m^3$), depth (0.44 $\pm$ 0.25 m) and eroded mass (61 $\pm$ 36 Mg). Despite their small sizes, they possess a significant impact on





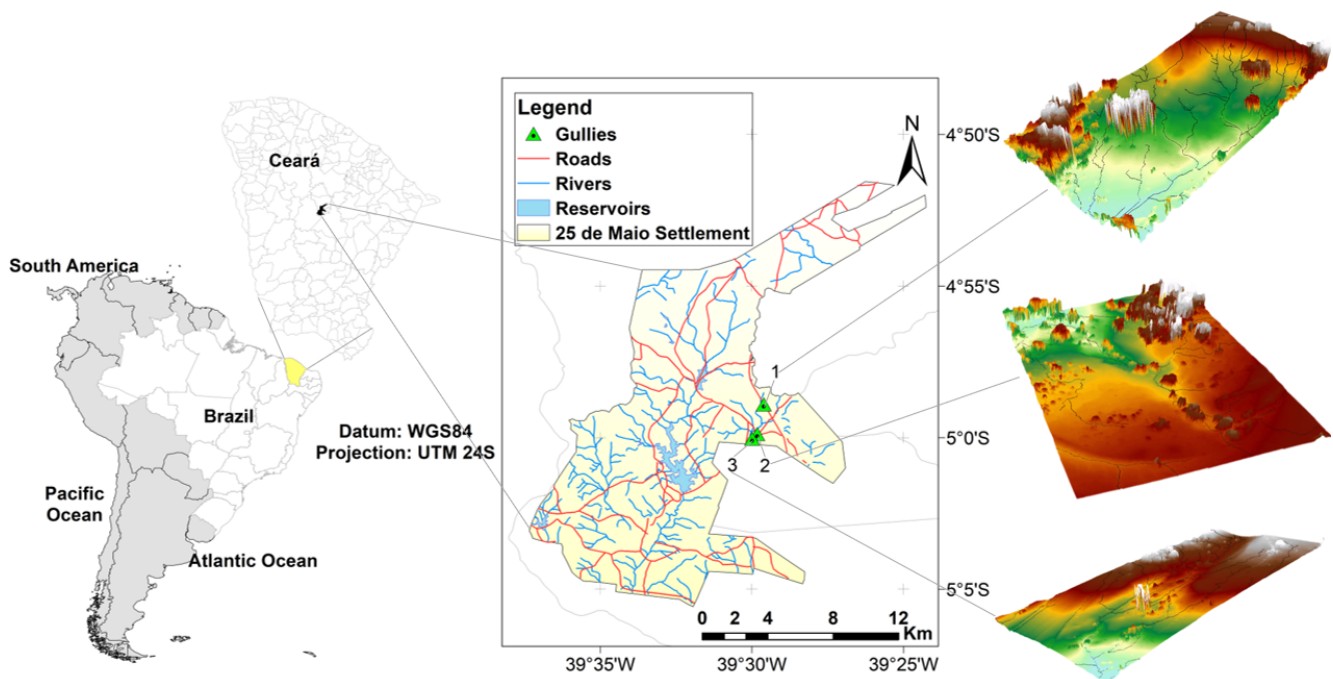

**Figure 1.** Location of the study area and the gully sites (gullies 1, 2 and 3) and the digital elevation models. The roads, rivers and reservoirs were mapped by Silva et al. (2015).

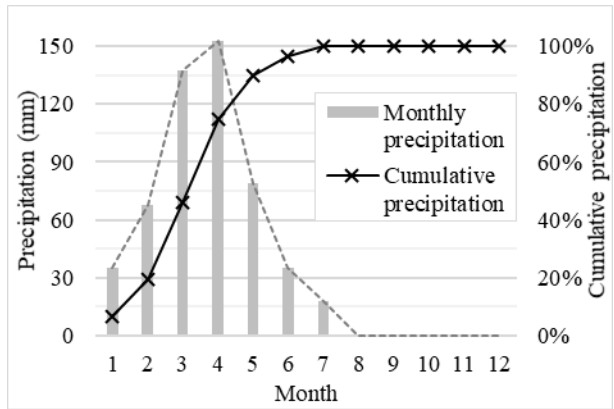

**Figure 2.** Monthly precipitation (median) and cumulative precipitation at MRB from 1958 to 2015.





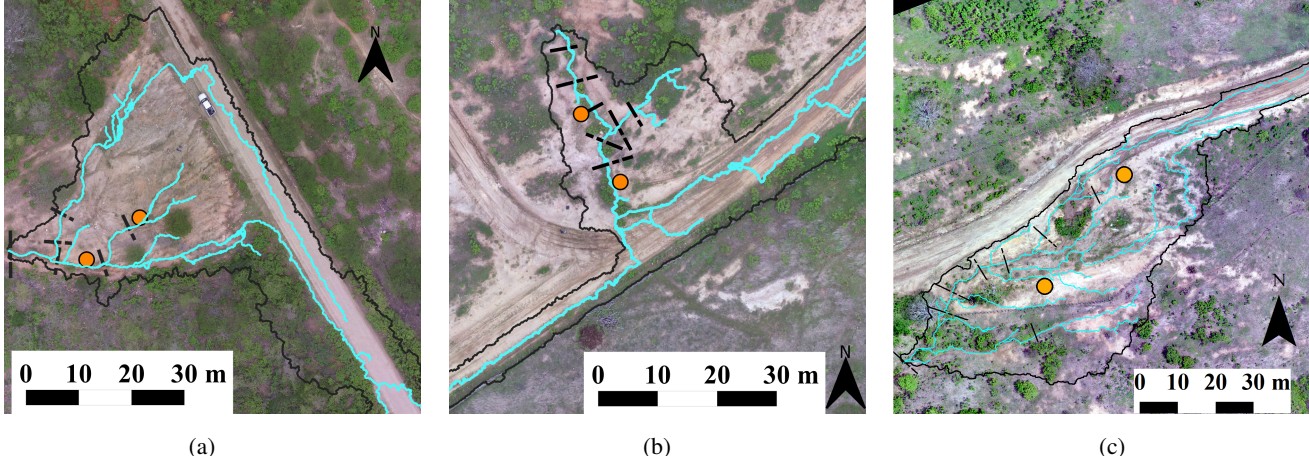

| (a) | (b) | (c) |

**Figure 3.** Aerial photogrammetry of the studied gullies. Note that they are at the margin of the road, receiving the concentrated flow diverged from it. The continuous black line represents the catchment boundaries; the blue line represents the flow paths; the dashed black lines are the cross-sections used on the validation of the model and; the orange dots are the soil sampling points - (a) gully 1; (b) gully 2; (c) gully 3.

the landscape for reducing productive areas and soil fertility. According to the information obtained from local villagers, gully erosion started immediately after the construction of a country road in 1958 (Figure 3). Before the construction, the sites were covered by Caatinga vegetation (Pinheiro et al., 2013). The road modified the natural drainage system and does not provide for any side nor outlet ditches, therefore generating a concentrated runoff at its side. This has caused excessive runoff on the hillslopes and triggered gully erosion.

## 2.2 Topography survey

The assessment of the gully data was achieved using an Unmanned Aerial Vehicle (UAV), a technique applied in other regions as well (Vinci et al., 2015; Wang et al., 2016; Pineux et al., 2017; Gudino-Elizondo et al., 2018). A UAV equipped with a 16 MP camera (4000 x 4000 pixels) and a view field of 94 % was used. The flight was at 50 m altitude with a frontal overlap of 80 % and lateral overlap of 60 %. For the geo-reference of the mosaic, five ground control points were deployed, which were evenly distributed in each area, both in high and low ground. The coordinates were collected using a stationary GNSS – RTK (L1/L2) system with centimetre-level accuracy. The Digital Surface Model was produced using the Structure from Motion technique. The process consists of a three-dimensional reconstruction of the surface, derived from images and the generation of a dense cloud of 3D points based on the matching pixels of different pictures and Ground Control Points (GPCs); the processing result is a model as accurate as one obtained by laser surveys (e.g., Light detection and ranging - LiDAR), but cheaper and less time consuming (Castillo et al., 2015; Agüera-Vega et al., 2017; Gudino-Elizondo et al., 2018). From the Digital Surface Model, the Digital Elevation Model (DEM) was derived. The ground sample distance (pixel size) obtained is of four to five centimetres and the digital models have high precision, with a vertical position error of around one centimetre and horizontal error of six





millimetres. The vegetation, yet sparse, was an obstacle to increasing the quality of the survey. However, as the focus of this
110  study was the gully cross-section geometry, vegetation interference was acceptably low.

### 2.3   Soil data

At the gully sites, soil surveys were carried out to assess the properties and parameters required to implement the model:
undisturbed soil samples were collected (see Figure 3) at depths of 0.10 m, 0.30 m and 0.50 m. At the depth of 0.50 m, a
well-defined horizon C, rich in rocks and soil under formation, was identified. The maximum depth of the non-erodible layer
115  ranged from 60 to 75 cm in all gullies. We performed grain-size distribution, sedimentation, organic matter, bulk density and
particle density analysis. Due to the scale of this experiment and the homogeneity of the soil-vegetation components (Güntner
and Bronstert, 2004) we divided the areas in two sets based on grain-size distribution, organic matter and bulk density. Gully
1 (G1) has specific features and comprises the first soil (S1), whereas gullies G2 and G3, close to each other, are represented
by the second soil (S2). The soils are loamy, with clay content ranging from 6 % to 37 %. The particle density is 2580 kg m$^{-3}$.
The soils are Luvisols and have typical profile, with the top layer relatively poor in clay when compared to the layers below
and with the regular occurrence of gravel at the surface. Furthermore, Luvisols are rich in active clay, which makes them prone
to form cracks and macropores when dry (dos Santos et al., 2016), a process also documented in soils with similar texture in
a semiarid area in Spain (van Schaik et al., 2014). Rill erodibility values (K$_r$) and critical shear stress ($\tau_c$) for the soils were
obtained using the Equations 1 and 2 (Alberts et al., 1995) and are also presented in Table 1.

$$K_r = 0.00197 + 0.00030\%VFS + 0.038633e^{(-1.84\%OM)} \tag{1}$$

$$\tau_c = 2.67 + 0.065\%C - 0.058\%VFS \tag{2}$$

where %VFS is the percentage of very fine sand, %C is the percentage of clay and %OM is the percentage of organic matter.

### 2.4   Rainfall data

Daily rainfall data for the location covering the entire period was provided by the Meteorological Service of Ceará (Funceme,
2018). We used of five rain-gauge stations in the region, covering the period from 1958 to 2015. The double mass method
was employed to check data consistency (Supplementary material - Fig. S1). The gaps in the measurements (January 1958 and
September 1960) were filled by the nearest gauging station. The annual rainfall in the area averages 613 mm (Supplementary
material - Fig. S2) and its coefficient of variation is 43%, typical values for the Brazilian Semiarid Region (de Araújo and
Piedra, 2009).

The modelling of the erosion processes is based on peak discharge, which demands sub-daily rainfall data, but only daily
precipitation is available inside the study basin. To accomplish the modelling, correlation curves were used, from the Aiuaba
Experimental Basin's (AEB) detailed hydrographs, which have been monitored since 2005 (Figueiredo et al., 2016). This





**Table 1.** Grain-size distribution, organic matter for both soils (S1 - for the gully 1 - and S2 - for the gullies 2 and 3) at three depths (10, 30 and 50 centimetres) and the respectives texture classification (USDA); and the estimated (in italic) rill erodibility ($K_r$) and critical shear stress ($\tau_c$ of the site soils) obtained using Equations 1 and 2.

| Soil and layer | Gravel > 2 mm | FCS[a] > 0.1 mm | VFS[b] > 0.05 mm | Silt > 0.002 mm | Clay < 0.002 mm | Organic Matter | Bulk density (kg m$^{-3}$) | Soil Class | $K_r$ (s.m$^{-1}$) | $\tau_c$ (Pa) |
|---|---|---|---|---|---|---|---|---|---|---|
| **S1-10** | 13 % | 45 % | 21 % | 11 % | 10 % | 3.1 % | 1699 | Sandy Loam | *0.015* | *2.102* |
| **S1-30** | 6 % | 46 % | 16 % | 14 % | 18 % | 3.3 % | 1677 | Sandy Loam | *0.016* | *2.912* |
| **S1-50** | 4 % | 63 % | 20 % | 7 % | 6 % | 2.2 % | 1765 | Loamy Sand | *0.020* | *1.900* |
| **S2-10** | 17 % | 33 % | 22 % | 11 % | 17 % | 4.9 % | 1509 | Sandy Loam | *0.012* | *2.499* |
| **S2-30** | 8 % | 29 % | 6 % | 20 % | 37 % | 5.7 % | 1572 | Clay Loam | *0.011* | *4.611* |
| **S2-50** | 2 % | 28 % | 15 % | 20 % | 25 % | 1.4 % | 1643 | Loam | *0.014* | *3.425* |

[a] Fine to Coarse Sand; [b] Very Fine Sand.

experimental basin is located 190 km south of MRB, but both basins are climatically homogeneous (Mendes, 2010). Figure 4 shows the rainfall data for the MRB collected during the rainy season in 2019 (January to July). We can observe that the data has similar behaviour but constantly on the lower area of the plot. It is relevant to note that the year of 2019 in MRB was dry and such behaviour is expected. To obtain discharge values from intensity we used the SCS-CN method (Chow, 1959). For the modelling effort, the main variables are the event peak discharge and its respective duration. Because the gully catchment areas are very small, their respective concentration time is negligible compared with the intense-rainfall duration in the region (Figueiredo et al., 2016), yielding a uniform pattern of peak discharge. For each event, four intensities were assessed: average ($I_{av}$), sixty-minute maximum ($I_{60}$), thirty-minute maximum ($I_{30}$) and fifteen-minute maximum ($I_{15}$) intensities. Figure 4 also depicts the correlation curves between daily precipitation and event duration, as well as those between daily precipitation and rainfall intensity at the Aiuaba Experimental Basin.

## 2.5 Coupled Model (FL-SM)

The proposed model, hereafter addressed as FL-SM, was designed to model small permanent gullies based on the coupling of the models of Foster and Lane (1983) and Sidorchuk (1999) further described bellow. The model proposed uses the equations of Foster and Lane for calculating the cross-section area and geometry over many events. The eroded layers, obtained for each erosive event, are piled and summed. While the Foster and Lane Model was applicable for single events generating rectangular-shaped sections, this model adaptation allows us to calculate the section geometry for long term gullies. The model inserts a routine for wall erosion based on the equations proposed by Sidorchuk, which are implemented once the gully reaches a threshold. This threshold is related to the cross-section area and catchment geometry. Once this threshold is attained, the equations of Sidorchuk are used for estimating the channel cross-section growth based on wall erosion. A flow chart of the model can be found in the supplementary material (Fig. S3).



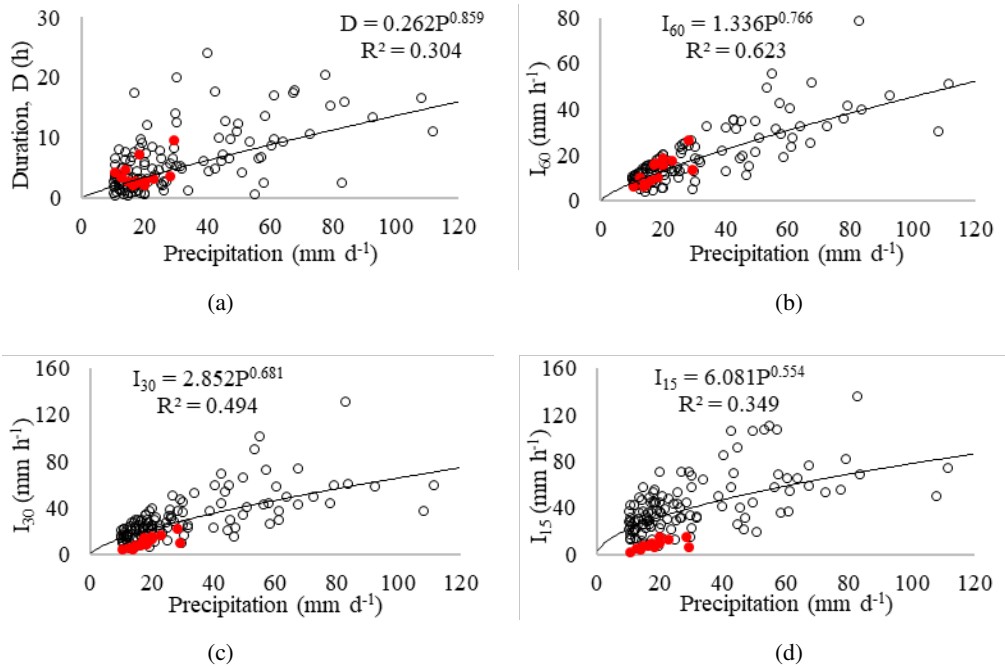

**Figure 4.** Correlation between daily precipitation and sub-daily variables at the Aiuaba Experimental Basin (Figueiredo et al., 2016). (a) daily precipitation versus event duration (D); (b) daily precipitation versus 60-minute maximum intensity ($I_{60}$); (c) daily precipitation versus 30-minute maximum intensity ($I_{30}$); and (d) daily precipitation versus 15-minute maximum intensity ($I_{15}$). The white circles indicate data obtained in Aiuaba from 2005 to 2014. The red dots indicate precipitations measured in the MRB from January to July 2019 (rainy season).

### 2.5.1 The Foster and Lane Model (FLM)

The Foster and Lane (1983) ephemeral-gully model aims at explaining "erosion by concentrated flow in farm fields" for single runoff intensive events. The gullies are considered ephemeral as productive farmlands usually provide periodic tillage to diminish or remove completely the gullies generated by previous events. The model is physically based, uses the Manning equation, mass balance, and shear stress mobilization; it assumes an equilibrium channel width and the gully evolution in two steps. The first step is the vertical incision, when the concentrated overflow starts digging the channel with a constant width.

The second step starts after the bottom of the channel reaches a non-erodible layer. Then, the section starts a sideward erosive process, widening until the end of the effective runoff, i.e., that with a shear stress below the critical stress. Detachment ratio ($D_r$) and shear stress ($\tau$) are given by the Equations (3) and (4).

$$D_r = K_r \left( \tau - \tau_c \right) \tag{3}$$

$$\tau(X) = 1.35 \gamma R_h S \left[ 1 - \left( 1 - 2 \frac{X}{WP} \right)^{2.9} \right] \tag{4}$$





In Equation 4, X is the position of a point on the channel bed, varying from zero to WP (wet perimeter). S is the longitudinal slope of the channel; $R_h$ the hydraulic radius; and $\gamma$ is the specific gravity of water (assumed 9.81 kN m$^{-3}$).

In order to be able to model long-term gullies using Foster and Lane (1983) equations, the following assumptions were made: first, all mobilized sediment is carried away by the discharges, i.e., there is no sediment deposition on the channel bed.

This assumption was confirmed by field surveys in many sections, where very little loose sediment was identified. Secondly, in the long run, the effect of each intense runoff can be piled in a cumulative model of widths/depths layers. This implies that each erosive event does not suffer significant influence of the previous, and the total eroded soil is related only with the energy of each event. The piling process considered all events with runoff. Flow charts for the original Foster and Lane Model and how it allows to model multiple events by piling area available in the supplementary material (Fig. S4 and S5).

To estimate the effect of wall erosion at the studied site, we first proposed an empirical parameter ($\lambda$ – Eq. 5) to correct the effect of lateral flow and wall erosion. This multiplicative parameter was calibrated and validated as function of the catchment shape, based on two coefficients: the Gravelius coefficient ($K_G$ – Eq. 6) and the form coefficient ($K_F$ – Eq. 7). Both coefficients describe the geometry of the catchment area and can be interpreted as how compact the area distribution is. Commonly linked to flood proneness, these parameters also relate to the transversal distance of the catchment area, which influences the amount

of lateral inflow into the mainstream. The diagrams of $\lambda$ versus both parameters are presented in Figure 5. Two equations [$\lambda(K_G)$ and $\lambda(K_F)$, see Figure 5] where calibrated using data from 14 randomly selected sections out of the 21 assessed by the DEM. The remaining data (from seven sections) were used to validate the equations.

$$\lambda = \frac{A_o}{A_m} \tag{5}$$

$$K_G = 0.28 \cdot \frac{C_P}{C_A} \tag{6}$$

$$K_F = \frac{C_A}{C_L^2} \tag{7}$$

In Equation 5 the terms $A_o$ and $A_m$ are the observed and measured cross-section area and in Equations 6 and 7, $C_P$, $C_A$ and $C_L$ stand for the catchment perimeter, area and length, respectively.

$$\lambda = \max\left(5.859 K_F^{0.707}; 1.0\right) \tag{8}$$

The coefficient ($K_F$) yielded a positive Nash-Sutcliffe Efficiency value and smaller RMSE (0.17 and 0.67 respectively),

which did not occur with the Gravelius coefficient (-2.43 and 0.84 respectively). In the revised model, hereafter addressed as FLM-$\lambda$, the FLM area output is multiplied by the calibrated parameter $\lambda$ (Equation 8), yielding the eroded area. Applying this factor caused a significant improvement in model efficiency, with NSE increasing from 0.557 to 0.757. The incremental area





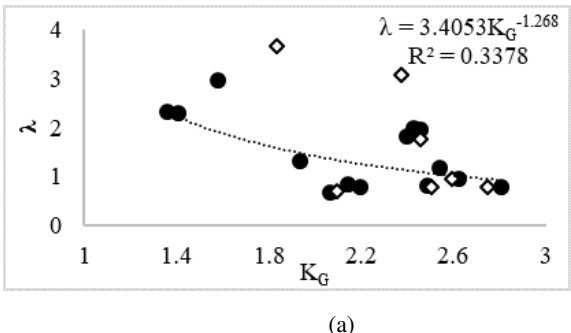
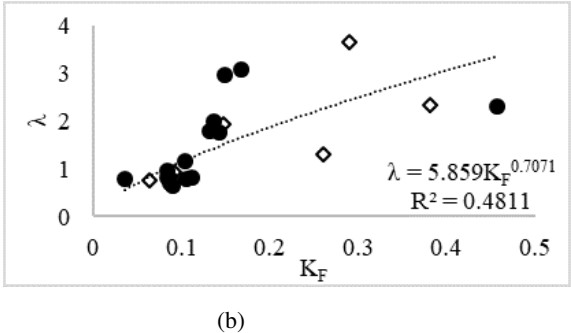

(a)                                                                                                (b)

**Figure 5.** Correlations between the ratio ($\lambda = A_o/A_m$) and (a) the Gravelius coefficient ($K_G$) and (b) the form factor ($K_F$) for 21 monitored cross-sections at MRB. Black dots refer to calibration cross-sections and white diamonds refer to validation cross-sections. The values of $R^2$ indicated in the plots are for the calibration. The validation $R^2$ were 0.10 for $K_G$ and 0.54 for $K_F$.

produced by the multiplication of $\lambda$ is assumed to increase the width of the upper half of the cross-section, keeping bottom width and the orthogonality of the walls unchanged.

### 2.5.2 The Sidorchuk Model (SM)

The Sidorchuk model (Sidorchuk, 1999) is physically and empirically based. It considers mass balance of sediment, shear stress (in terms of critical velocity), soil cohesion and the Manning equation to estimate the cross-section geometry and channel slope. It also uses empirical equations based on field measurement to estimate the flow depth and width. The model gives special attention to the processes involving gully wall transformation, as shown in Equations (9) and (10).

$$D_{vcr} = \frac{2C_h}{g\,\rho_s} \cos(\varphi)\sin^{-2}\left[\frac{1}{2}\left(\varphi + \frac{\pi}{2}\right)\right] \tag{9}$$

$$\frac{C_h}{g\,\rho_s\,D_v} = \frac{\rho_s - w\rho}{\rho}\tan\varphi\cos^2\phi - \frac{\sin 2\phi}{2} \tag{10}$$

In Equations (9) and (10), $C_h$ is soil cohesion (MPa); $D_v$ the depth incision (m); $D_{vcr}$ the critical value of depth for wall failure; $w$ is the volumetric soil water content; $\rho_s$ is the bulk density of the soil; $\rho$ is the density of water; g is the gravity acceleration; $\varphi$ is the soil internal friction angle; and $\phi$ is the wall slope, in degrees.

### 2.5.3 Threshold definition

The FL-SM requires the determination of a threshold value for the implementation of the wall erosion equations. Such a threshold controls when the wall erosion becomes significant for the total amount of eroded soil. In the model, it represents the limit stage, above which Sidorchuk (1999) equations are used. It also represents the scale when solely the channel bed erosion,





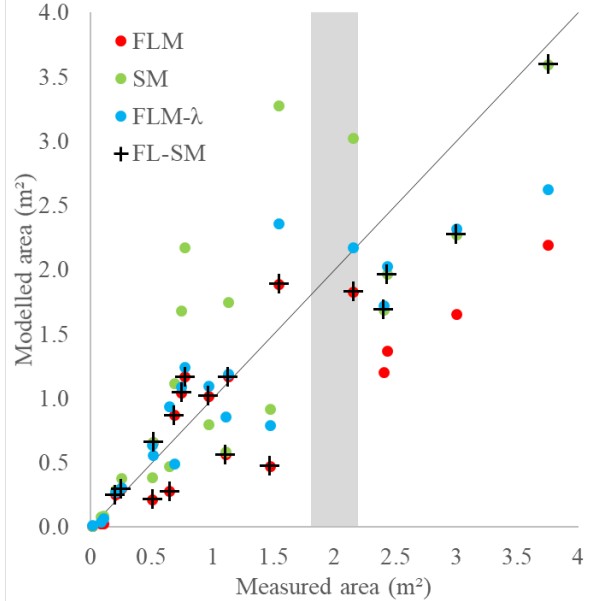

**Figure 6.** Performance of the coupled model (FL-SM), Foster and Lane Model (FLM and FLM-$\lambda$) and the Sidorchuk model (SM).

described by the Foster and Lane (1983) equations start to consistently underestimate the measured area. In this study we used

the Foster and Lane model to identify this scale where the both processes (channel bed and wall erosion) switch relevance.

### 2.6    Model fitness evaluators

To assess the goodness-of-fit, the Nash and Sutcliffe (1970) efficiency coefficient (NSE); the root mean square error (RMSE); and the percent bias (PBIAS) were used (see Moriasi et al. (2007)). Besides, the methodology proposed by Ritter and Muñoz-Carpena (2013) asserts statistical significance to the evaluators. The proposed model is based on Monte Carlo sample tech-

niques to reduce subjectivity when assessing the goodness-of-fit of models.

### 3    RESULTS

### 3.1    Coupled model (FL-SM)

The FL-SM presented a Nash-Sutcliffe Efficiency coefficient of 0.846 when using a threshold for the area of the cross-section of 2.2 m$^2$. This value can be slightly improved if a parameter $\lambda$ is calibrated for sections that are smaller than the threshold

(FL-SM-$\lambda$), yielding a NSE of 0.853. fBy doing this, however, a calibrated parameter has to be reintroduced, which is not desired. Figure 6 presents the scatter of modelled and measured data for the models implemented. In Figure 7 some output examples for the sections above the threshold are presented.





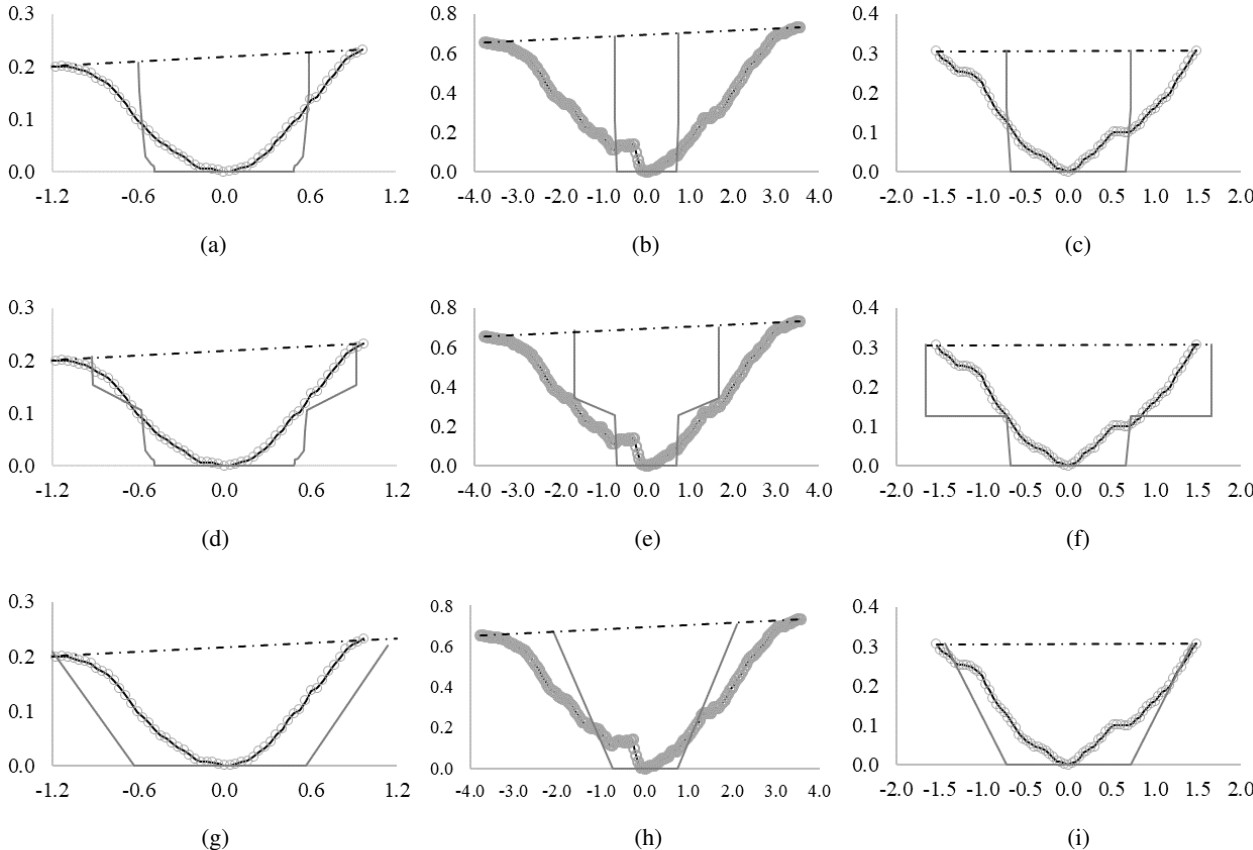

**Figure 7.** Some examples of gully cross-sections measured (black line with circles) and the modelled (dark grey line) geometry. figures (a), (b) and (c) show the output for the model of Foster and Lane; figures (d), (e) and (f) the output for the model of Foster and Lane adapted with the parameter $\lambda$ and figures (g), (h) and (i) the result from the Sidorchuk Model (SM). Distances in metres. Section in (a, d and g) is a section obtained from gully 1, (b, e and h) from gully 2, and (c, f and i) from gully 3.

In terms of geometry, sections below the threshold have cross-sections similar to Figure 7 [(a), (b), (c)], with rectangular-like shape, unless the $\lambda$ parameter is reintroduced, which leads to cross-sections like Figure 7 [(d), (e), (f)], with piled rectangles.
When the area surpasses the threshold value, sections have mainly trapezoidal geometry, as illustrated in Figure 7 [(g), (h), (i)]. It is important to highlight that the model can produce triangular geometry, but none was obtained in this study.

### 3.1.1 Threshold analysis

The interpretation of the threshold for implementation of the wall erosion routine can be based on (a) the cross-section area or (b) the catchment geometry, as illustrated in Figure 8. The first approach considers a critical area that once reached marks
when the wall erosion is really significant over the other processes. In this study the threshold identified was at an area of 2.2 m$^2$. After that, the model calculates the effect of sidewall erosion and reaches the critical final area for the analysed section.





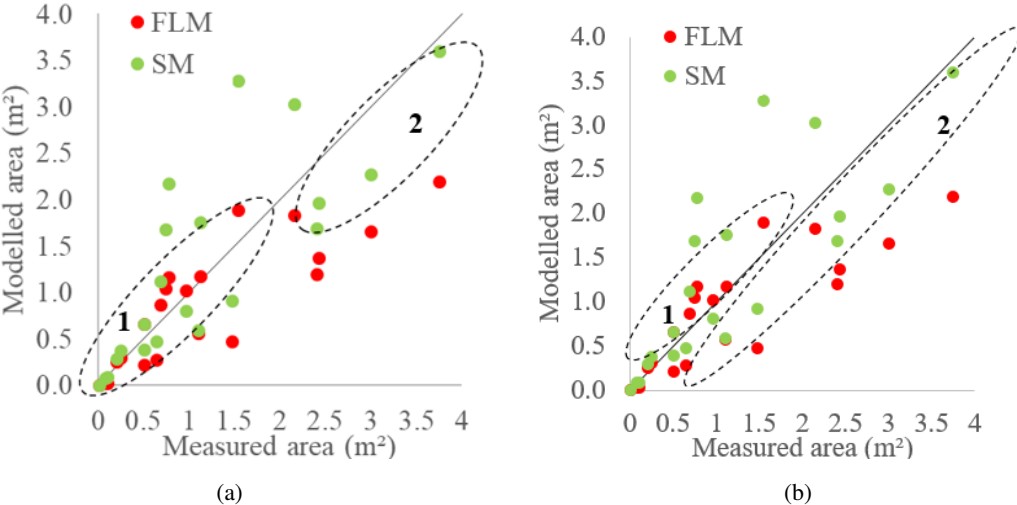

(a)                                    (b)

**Figure 8.** Thresholds for wall erosion: (a) based on the cross-section area; (b) based on the catchment geometry and $K_F$. In both plots the set 1 indicates the domain of bed erosion and Foster and Lane equations and set 2 the domain of wall erosion and Sidorchuk equations.

The presence of a threshold for applying the sidewall erosion routine indicates a change of relevance among processes on a given scale. Although the threshold is addressed as an area, this is only a consequence of more complex interactions among discharge, flow erosivity, cohesion and gravitational forces.

The second interpretation is related to the catchment geometry, as the approach given to the parameter $\lambda$ also related to the $K_F$. From the distribution of the cross sections we can observe sections that are better modelled by SM even under the threshold. By analysing the values of form coefficient ($K_F$) of each set (Figure 8b) we observed that set 1 has $K_F$ of ($0.08 \pm 0.02$) and set 2 has $0.22$ ($\pm\ 0.12$). Higher values of $K_F$ indicate a more compact catchment, with more lateral flux into the channel, therefore producing more erosion in the soil. By sorting the model results of FLM and SM based in the form coefficient, using

the threshold of $K_F = 0.15$, we obtained an NSE of 0.79.

### 3.1.2   Rainfall intensity

Of the three gullies, twenty-one cross sections with no interference of bushes or trees were selected from the Digital Elevation Model. The FLM was tested for the 60-minute, 30-minute, 15-minute and average intensities [FLM($I_{60}$), FLM($I_{30}$), FLM($I_{15}$), FLM($I_{av}$)]. It showed the best response when using the thirty-minute intensity [FLM($I_{30}$); NSE = 0.557]. Figure 9 presents

the plot of the model outputs for area and width compared with measured data. One can observe that no tested intensity gave a realistic approach to the observed values of width. Moreover, the model did not show good responses to the cross-section geometry, regardless of the intensity tested. This may indicate a flaw of the model concerning the sideward erosive process. To enhance model performance, the focus was set on cross-section erosive processes. The FLM considers a rectangular shape for the sections, but the field survey showed that the sections are rather trapezoidal or triangularly shaped (Figure 10). Among the





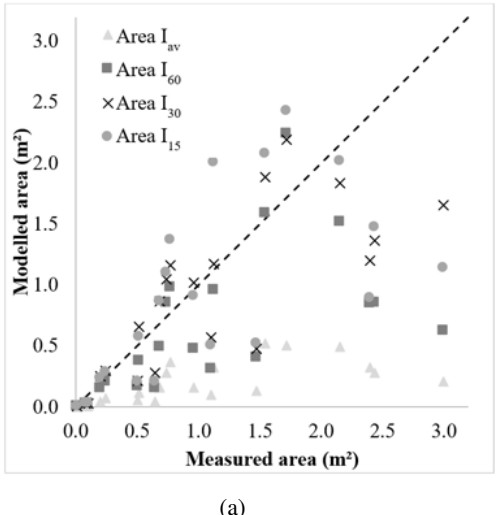 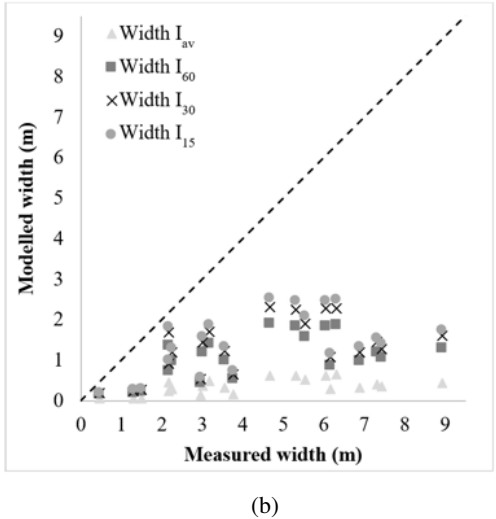

(a)                 (b)

**Figure 9.** Results of the FLM for the different intensities ($I_{av}$ – average; $I_{60}$ – 60-min; $I_{30}$ – 30-min; and $I_{15}$ – 15-min) to generate the peak discharge. Scatter of points comparing measured and modelled values: (a) Area; (b) Top Width.

factors that can shape gully walls, there are seepage, slope material, and the slope angle itself (Bocco, 1991; Sidorchuk, 1999; Martínez-Casasnovas et al., 2004; Bingner et al., 2016). Besides, gully walls can be shaped by lateral discharge (Blong and Veness, 1982), which depends directly on the morphology of the cross-section catchment area. Figure 9 also shows a tendency of the model in underestimating the cross-section area, which implies that the model does not consider all the relevant erosive processes. The sidewall erosion has proven to be a relevant source material, often representing over 50 % of the eroded mass

(Crouch, 1987, 1990), whereas the FLM only assumes the vertical sidewall morphology. Hereafter the acronym FLM will refer solely to the model implemented with 30-minute intensity.

### 3.2   Model evaluators

The coupled model, FL-SM, presented the highest performance of goodness-of-fit evaluators (Figure 10a). The model yielded a PBIAS value below 10 %, which is very good. The coupled-model RMSE was also low (1.82), whereas the NSE reached a

value of 0.846, being classified as good (Ritter and Muñoz-Carpena, 2013) or very good (Moriasi et al., 2007).

     Figure 10b shows the evolution of NSE values with more details so that conclusions can be drawn. The Foster and Lane model with $\lambda$ parameter (FLM-$\lambda$) was calibrated with 14 cross-sections out of 21 and performed as well as the Sidorchuk model (SM), which considers the sidewall effect. For the coupled model, there is no efficiency gain when applying the calibrated parameter ($\lambda$) to sections below threshold which indicates that the lateral inflow is only relevant for larger sections.

Another approach to estimate the goodness-of-fit of a model was proposed by Ritter and Muñoz-Carpena (2013). The routine (FITEVAL) consists of repeatedly resampling from the dataset and handles each resample as an actual sample of the population. This grants the generation of a confidence interval for statistical evaluators. The method classifies the model as acceptable to





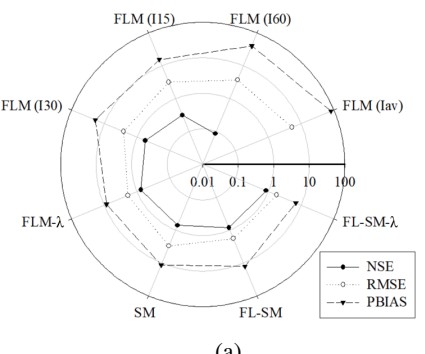
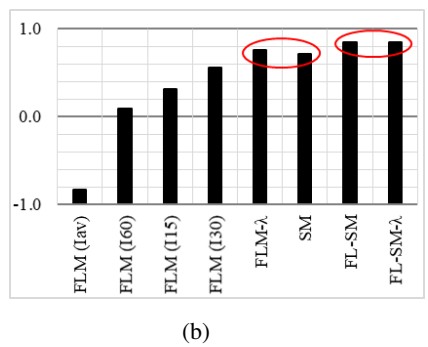

(a)  (b)

**Figure 10.** (a) Model evaluators NSE, RMSE and PBIAS. The web graphs shows the performance of all tested models – values of PBIAS in percentage. (b) Evolution of the NSE values for every model. The red balloons indicate models with similar efficiency. Values of PBIAS in percentage.

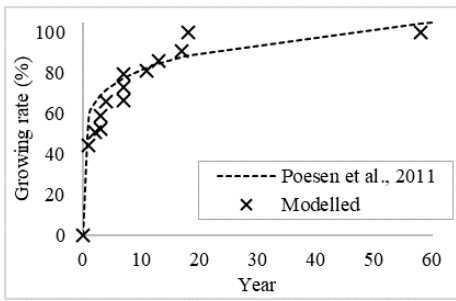

**Figure 11.** Behaviour of gully growing rate as proposed by literature (Poesen et al., 2011) and modelled (data from Gully 1).

very good – NSE $\in$ [0.66;0.95] for a p-value of 0.05. A conservative interpretation of this result implies considering the lowest values as the minimum state of information, or as the one that contains (almost) no unproven hypothesis. As a consequence, and according to Ritter and Muñoz-Carpena (2013), the FL-SM can be classified as acceptable to very good. The detailed output of the FITEVAL analysis can be found in the supplementary material (Fig. S6).

### 3.3 Gully growing modelling

The model also mimics the growing dynamic of gullies and its latency periods between extreme events, as reported in literature (Vanwalleghem et al., 2005; Poesen et al., 2011; Poesen, 2018) and illustrated in Fig. 11. Gully growing is commonly described as being a fast process in the first years and then progressively slowing down its enlargement. In our model, the mechanism that produces this dynamic is event piling. It could be observed that after a particularly intense event, the channel is sufficiently wide, whereas less intense events produce only a shallow flow with low shear stress and, therefore, no erosion. Only when a more intense event then the last erosive one happens, there is erosion. A flow chart of the final model (Foster-Lane-Sidorchuk Model) is available in the supplementary material (Fig. S5).





## 3.4 Landscape development impacts on gully erosion

Gullies are scale-dependent phenomena and frequently related to thresholds due to their initiation, which is based on catchment area and slope (Poesen et al., 2003; Torri and Borselli, 2003; Torri and Poesen, 2014; Poesen, 2018). Both characteristics are directly linked to shear stress and stream power when using physical gully models. Montgomery and Dietrich (1992) argue that changes in landscape and the drainage system can lead to a larger occurrence of channelization and its impacts can be noticed faster. Torri and Poesen (2014) suggest a threshold for head development in gullies as conveyed in Equation 11.

$$S\, C_A^{0.38} > k \tag{11}$$

where S (m m$^{-1}$) is the slope, $C_A$ (ha) is the catchment area and $k$ in a parameter for channel and gully initiation.

For croplands in tropical conditions, the proposed value of $k$ is 0.042 (Torri and Poesen, 2014). For the gullies in the present study, however, a head development for values lower than half ($k$ = 0.020) and systematically lower than the field data of (Vandaele et al., 1996) could be observed. These findings suggest soil vulnerability of the gully erosion region. Considering that the three experimental sites were located next to a road, this disturbance triggered gully initiation and other actions may cause similar problems in the region, such as deforestation and forest fire. The road not only enlarged the total catchment area, it also increased its length. While relations between catchment length and area are well-established (L = b $C_A^{0.49}$) with values of b varying from 1.78 to 2.02 (Montgomery and Dietrich, 1992; Sassolas-Serrayet et al., 2018), the present experiments found b equalling 3.17. With a smoother surface and almost no meandering, road construction caused modifications that promoted more energetic flows on the gully head. Road construction has also been identified as potential factor for gully initiation in other areas of the Brazilian Semiarid Region, as in the Salitre Catchment, where large gullies started after construction of an unpaved rural road (Souza et al., 2016; da Silva and Rios, 2018).

## 4 DISCUSSION

### 4.1 Coupled model (FL-SM)

The FL-SM considers two sediment sources, channel bed and sidewall. Gullies are, however, complex systems with many sources and interactions. Headcut, sidewall erosion due to raindrops and piping were not considered in our modelling approach. Processes of infiltration, subsurface flow and transport capacity were also neglected and should be properly addressed in future works. Nevertheless, the FL-SM assumptions managed to mimic well the field measurements, which implies that, at least in this study, the neglected processes are of lower relevance or were considered indirectly. For instance, sidewall erosion by raindrops can be considered insignificant over the wall failure process considered by Sidorchuk (1999). In addition, it is important to notice that, by selecting the 30-minute intensity, a less intense interval might be overlooked that produces erosive discharge, too, and can, therefore, explain the remaining processes.





Despite the good results obtained from the modelling, the use of stochastic approaches (Knapen et al., 2007; Sidorchuk,
2005) should improve the performance of the model and introduction of other sources of sediment. This is also relevant for
generalisation and modelling of classical gullies. In the same way, the introduction of processes as armouring and energy losses
as proposed by Hairsine and Rose (1992) can be interpreted as probabilistic terms.

### 4.1.1  Foster and Lane Model (FLM)

The first attempt for this study was to identify which was the best peak and duration of rainfall to be considered. Due to the
timescale of this work, it was required to select a single parameter for all the events. Therefore, a relevant result from this work
is the confirmation that the 30-minute intensity is the one that provides most information about gully erosion. Wischmeier and
Smith (1978) proposed the product of total storm energy and the 30-minute intensity to "predict the long-time average soil loss
in runoff". The use of $I_{30}$ for estimating event-related gully erosion was previously experimentally tested by Han et al. (2017).
The authors had monitored a gully in the Loess Plateau in China for 12 years, registering 115 erosive rainfall events. They
concluded that the product of 30-minute intensity and total precipitation (P $I_{30}$) was the key parameter to estimate total soil
loss. Our results corroborate with this.

Furthermore, by applying the $I_{30}$ in this study in order to estimate peak discharge and duration, it is implied that all the
energy necessary to initiate and develop a gully channel comes from the most intense 30 minutes. Due to the limited number
of gullies, it is not straightforward that the $I_{30}$ could be the most representative index for any situation. Peak discharge and
critical rainfall duration are often central variables in gully models (Foster and Lane, 1983; Watson and Laflen, 1986; Hairsine
and Rose, 1992; Sidorchuk, 1999; Gordon et al., 2007) and are related to erosion initiation parameters and thresholds, such as
shear stress and stream power. This second factor has frequently been reported in literature as being more correlated with both
laminar and linear erosion (Torri and Borselli, 2003; de Araújo, 2007; Nazari Samani et al., 2016; Bennett and Wells, 2019).
Figure 9a shows the performance of the tested intensities. Although the model using the 15-minute intensity presented smaller
PBIAS and RMSE, the results indicated a large scatter around average. The FLM was further improved by the addition of the
calibrated parameter $\lambda$ (FLM- $\lambda$). This parameter is included to predict the effect of lateral discharges over wall erosion. Due
to the significant improvement produced by its insertion, it could be understood that the original FLM fails to tackle this source
of material (Blong and Veness, 1982; Crouch, 1987).

### 4.1.2  Sidorchuk Model (SM)

The SM produced good results in this study, which were similar to those obtained by inserting a calibrated factor ($\lambda$) in FLM.
It is important to note that the original model used empirical correlations to determine width (Sidorchuk, 1999; Nachtergaele
et al., 2002) and those were obtained for the Yamal basin. In the present study, we substituted this approximation for the width
estimated by the FLM model, which permitted a more physical approach and increased the quality of the SM. The model was
also capable to predict well the sidewall slope.

The model, however, showed a trend of overestimating smaller cross-sections (see Figure 6), mainly due to section geometry.
When applied, the bottom width of the channel is considered to be the final width obtained by FLM. In larger sections, this





hypothesis holds, once the discharge is large enough to carry all soil produced by sidewall erosion. In smaller sections, part of the soil is deposited and produces a V or U shaped cross-section (Starkel, 2011).

## 4.2 Topographic data

In terms of accuracy and agility, a topographic survey with UAV exceeds the Digital Elevation Model and permits to measure sites within a few minutes. Conventional measurements, such as those with total station or profilometer, are more time consuming and do not grant better resolutions. The UAV accuracy, however, can be enhanced by performing flights in lower heights and more GCPs (Agüera-Vega et al., 2017; James et al., 2017), as well as by using high-end equipment, such as more robust UAVs and stabilizers. Total stations can also be used to improve the accuracy of ground control points (Mesas-Carrascosa
et al., 2016). Given the scale of this study and the presented results of the models, the four-centimetre pixel represents a good resolution, since it combines good precision with affordable computational costs (Wang et al., 2016). The solution of UAV-based volume assessment is a good option for monitoring gully evolution (Marzolff and Poesen, 2009; Stöcker et al., 2015), allowing frequent surveys, e.g., after every intense rainfall event. Trees and bushes obscure topographic measurements if too close to the gully channel and/or too dense in the catchment. Thus, UAV monitoring is more reliable for gully sites in non-
or meagre-vegetated areas and meadows, which combines with the conditions of this study, except for gully 3, where it was impossible to accurately measure the total erosion volume due to relatively dense vegetation. It was, however, possible to select a large enough number of sections (eight) at gully 3 to assess the total erosion volume. The topographic survey shows that the gully watersheds contain large parts in the road, indicating a modification of the drainage system and change of the catchment boundaries – both causes of gully initiation foreseen by Ireland et al. (1939) – due to road construction, which promoted in-
tense runoff and triggered gullies. Impacts of road construction on gully formation were also observed in Ethiopia (Nyssen et al., 2002) and the USA (Katz et al., 2014). Considering such previous records in literature and the information collected with locals, the modelling considered 1958 as the start of gully erosion, coinciding with road construction.

## 4.3 Soil data

Though the three studied gullies are located in the same mesoscale basin, the Caatinga biome is known for its soil variability
(Güntner and Bronstert, 2004) and soil properties do differ among the gullies. However, only small changes of texture were observed in different depths, allowing an analysis based on average properties. Nevertheless, for deeper and/or more variable soils, the discretization of soil properties, and therefore parameters such as rill erodibility ($K_r$) and critical shear stress ($\tau_c$) can easily be taken into account. The good performance of the final model (FL-SM) also indicates that the WEPP equations for critical shear stress and rill erodibility (Eq. 1 and 2) can be used for the soils of the region. These equations were obtained
via regression curves from data collected on 34 plot areas in the USA with a wide range of textures, slopes, land use and land cover. The areas from the WEPP model possess different geological and climatological conditions from the soils in the Brazilian Semiarid Region; this is why local studies should be carried out, since empirical equations frequently have strong local character (Ghorbani-Dashtaki et al., 2016; Dionizio and Costa, 2019).





### 4.4 Rainfall data

This study shows that sub daily information of rainfall is of crucial importance for gully modelling. In this study, we used correlation curves based on long-term time series of a similar catchment in the region. However, such analysis might introduce an averaged and monotonic behaviour for the intensities, as illustrated in Figure 4, and is, therefore, unrealistic. Stochastic models should be tested to estimate sub daily information from daily rainfall. The estimation of discharge from rainfall can also be improved by considering water content in the soil and modelling its evolution over the studied period using water

balance models.

## 5 Conclusions

In this study, efforts were concentrated on understanding how the cross-section of small permanent gullies (with no groundwater contributions) evolve. It was possible to identify two main mechanisms: the first is bed erosion, governed by shear stress at the bottom of the channel. The second is the sidewall erosion due to gravitational processes and lateral flow. To successfully model

both components, two models were coupled – those by Foster and Lane (1983) and Sidorchuk (1999). The two mechanisms, however, do not happen simultaneously in all sections, they depend on the process scale and are ruled by a threshold of eroded cross-section area. To model the gullies in a region without local data of sub-daily rainfall, correlation curves were used to estimate rain intensity from total daily precipitation. For the purpose of modelling peak discharges, the intensity that showed the best results was the 30-minute one($I_{30}$).

Gully is an erosion related to many processes and it is scale dependent. The attempt of proposing a generalist model for gullies should also consider these different scales and mechanisms involved in different stages of the gully development. Catchment shape and lateral flow have a central role on gully erosion and their influence should be further investigated. Infrastructure constructions, as roads, change conditions for gully initiation and was the trigger for the studied gullies.

The proposed final model (FL-SM; standing for Foster & Lane and Sidorchuk Model) presented a performance that varied

from acceptable to very good, depending on the criteria. This model managed to estimate total erosion in a small permanent gully for a total period of six decades. Despite the good results of FL-SM, more efforts should be undertaken to tackle other sources of gully sediments, such as headcut and piping..

*Code and data availability.* Code and data available in the link: https://github.com/PedroAlencarTUB/GullyModel-FLSM

*Author contributions.* Alencar worked on fieldwork, programming, laboratory analysis. Teixeira carried image acquisition and processing.

de Araújo acted as supervisor of the work and in its conceptualization. Alencar prepared the text with contributions of all authors.



*Competing interests.* The authors declare that they have no conflict of interest.

This work is part of the PhD work of **Pedro Alencar** and will be used in his dissertation.

*Acknowledgements.* This study was partly financed by the *Coordenação de Aperfeiçoamento de Pessoal de Nível Superior - Brasil* (CAPES)
- Finance Code 001 and by the *Edital Universal* CNPq - number 407999/2016-7. Pedro Alencar is funded by the DAAD. Many thanks to
Prof. Eva Paton for the support and advising at the Technische Universität Berlin.



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
