# Peer review of "Physically-based model for gully simulation: Application to the Brazilian Semiarid Region"

_Hydrology and Earth System Sciences, 2019_

## Referee Comment (RC1) · Anonymous Referee #1 · 25 Jan 2020

Pedro Henrique Lima Alencar et al. 2019, entitled "Physically-based model for gully simulation: application to the Brazilian Semiarid Region," studies the applicability of two simplified/empirical or physically based erosion models for a study area in the semiarid region of Brazil. The author's choice of the study area is commendable as almost no attention would be paid whatsoever in such small gullies those are erosion susceptible. The author's performed properly planned investigations, i.e., topographic survey and soil data collection. Two models are compared i.e., FL and SM and the authors propose a combined modelling approach which they name it as FL-SM model. The authors are familiar with the codes and have used them for their study efficiently. The model evaluation performed shows the reliability of their approach. I commend

the number of efforts put by the authors. However, I do have some major significant suggestions which would be helpful for the authors to improve the presentation of their data and readability of the manuscript. I would suggest major revisions, and after substantial revisions, this manuscript would be a useful contribution. Specific comments: 1. Abstract: Though the abstract tries to suffice most of the stuff from the manuscript, the way it is written makes it very hard to read and understand. A complete rewriting of the abstract is required. For example, the first two sentences could be written as "Gullies are most prone to erosion processes, leading to land degradation and desertification, especially in arid and semiarid regions. The second sentence could be written as "despite the slowly possessed threat of gully erosion, there are not many developments being made in this regard." 2. Introduction: Similar to the abstract, the introduction part should also be revisited. The authors refer to the global and local scale importance of gully erosion by providing many examples but do not address how and why gully erosion is an issue in their selected study area. Further, the main contents of the manuscripts, the objectives and methodology part are not summarised in the introduction. 3. Coupled Model (FL-SM): The authors describe the governing equations of the FL model and SM model separately. It is not very clear how these equations are coupled and what platform the authors have used to run their simulation. The flow-chart showed in supplementary Fig. S3 should be moved to the main text and the modifications performed by the authors should be shown more clearly so as to ease the understanding of readers. Without that, it looks like the authors did not perform substantial modifications of the code. The evaluation of the proposed coupled model (FL-SM), along with the FL and SM models shown in Fig 6, doesn't tell the quantitative performance. I suggest the authors add the $R^2$ value for each model and verify whether a statistically significant result is gained (p-value test) or not. The same can be done for Fig 8 and Fig 11. It is also not clear why the authors have shown the rainfall comparison plots within the modelling section. Further, the explanation of Fig 5, and its relevancy to the corresponding section (modelling) is not understandable. 4. Model evaluations: The authors have put a lot of effort into validating their new model. But it

is very difficult to understand the model evaluator shown in Figure 10. Further explanations would suffice. The evaluation for the gully growing modelling provides satisfactory results. It would be better if the authors explain what methods did Poesen et al. (2011) employ. 5. Discussion part: A long discussion is provided explaining the limitations of the model and data availability, i.e., topographic, soil and rainfall data. The titles of the sub-sections could be rethought as 4.1. model limitations and 4.2 data limitations. 6. English language usage: Though the authors have put a lot of effort into the technical aspects of this manuscript, the overall writing could be improved much. I suggest the authors have a native speaker to check and rewrite the manuscript. Technical corrections: 1. Most of the figures are referred to at places after the figure is presented. The flow of the overall reading is not smooth as many figures are misplaced. 2. The authors are requested to first site the figure in the manuscript and then put it somewhere near. Also, figures are pasted in different sections, which is not relevant. 3. Overall, I would suggest the authors perform professional proofreading and grammar corrections. 4. The supplementary files just contain some figures without any description regarding how to use them whatsoever. 5. In my opinion, this manuscript does not have major technical flaws despite the weak overall structure and complex writing style.

---

## Referee Comment (RC2) · Anonymous Referee #2 · 23 Feb 2020

Physically-based model for gully simulation: application to the Brazilian Semiarid Region

Dear Editor I have went through the above article. I think there are the following comments before making an acceptance decision. At present there are moderate to major comments. My decision is Major correction. Best

Manuscript need to check by a native for removing some language errors.

Abstract I cant see any quantitative results in abstract. This part need to write again. 1 Introduction Lines 16-17: this sentence needs to reference. There are too many citation in this part. Please decrease them and try to just use from newest citations. Please

clearly specify innovative of the current study 2.1 Study area Please add geographical coordinate of the study area. Lines 83-84: please add a reference. The same lines 85-86. Figure 1. I cant see coordinate system on maps. 2.3 Soil data How many sample do you use in this study? Figure 4. Correlations are so low. Why? What is its reason? 2.5.1 The Foster and Lane Model (FLM) and 2.5.2 The Sidorchuk Model (SM): These are routine and readers could find them in literatures. You have to present your combined model very carefully. This is so important. Please edit this part.

2.6 Model fitness evaluators Please add equations of the used evaluation methods. Results This part is written very carefully. Discussion I think it is important to add some discussions and comparisons by previous works. 5 Conclusions This part is so general. Please add some suggestion on this new model and etc. . .

---

## Author Comment (AC1) · 16 Mar 2020

**Answer to comments of Anonymous Referee #1**

We would like to sincerely thank the Referee for his/her insightful comments about our work and all the time spent on our manuscript. All observations and suggestions helped us to improve our work, for which we are grateful.

The main points highlighted by the Referee concern **Writing structure, Model structure** and **Statistical analysis of model performance.** Below we present a point-by-point answer to each topic.

**1. Abstract:** Though the abstract tries to suffice most of the stuff from the manuscript, the way it is written makes it very hard to read and understand. A complete rewriting of the abstract is required. For example, the first two sentences could be written as "Gullies are most prone to erosion processes, leading to land degradation and desertification, especially in arid and semiarid regions. The second sentence could be written as "despite the slowly possessed threat of gully erosion, there are not many developments being made in this regard."

Thank you for the observations and rephrasing suggestions. The Abstract was rewritten in order to make it clearer. In the box below we present the new abstract

Gullies lead to land degradation and desertification, an increasing environmental and societal threat especially in arid and semiarid regions, despite of which there is a lack of research initiatives in this regard. As an effort to better understand soil loss in those systems, we studied small permanent gullies, a recurrent problem in the Brazilian North-eastern semiarid region. The increase of sediment connectivity and reduction of soil moisture, among other deleterious consequences, endangers this desertification-prone region and reduces its capacity to support life and economic activities. Hereafter, we propose a model to simulate gully-erosion dynamics, derived from the previous physically-based models by Foster and Lane and by Sidorchuk. The models were adapted so as to simulate long-term erosion. A threshold area shows the scale dependency of gully erosion internal processes (bed scouring and wall erosion). To validate the model, we used three gullies ageing over six decades in an agricultural basin in the State of Ceará. The geometry of the channels was assessed using UAV (Unmanned Aerial Vehicle) and Structure-from-Motion technique. Laboratory analyses to obtain soil properties were performed. Local and regional rainfall data were gauged to obtain sub daily rainfall intensities. The threshold value (cross-section area of 2 m$^2$) characterise when erosion in the walls due to loss of stability becomes more significant than the detachment of sediments in the wet perimeter. The 30-minute intensity can be used when no complete hydrographs from the rainfalls are available. Our model can satisfactorily simulate the gully-channel cross-section area growth over time, yielding Nash efficiency of 0.85 and R$^2$ of 0.94.

**2. Introduction:** Similar to the abstract, the introduction part should also be revisited. The authors refer to the global and local scale importance of gully erosion by providing many examples but do not address how and why gully erosion is an issue in their selected study area. Further, the main contents of the manuscripts, the objectives and methodology part are not summarised in the introduction.

A revision of the introduction will be performed both in terms of structure and language.

- One paragraph was included after L54 introducing the area of study and the impacts of gullies in the region:

*The State of Ceará, located in the semiarid region, has its total area (over 148000 km²) included in the risk zone of desertification. From this total, about 11.5 % is also under advanced land-degradation conditions, including the formation of Badlands and Gullies, a similar condition to the one found in other desertification hotspots in semiarid regions (Mutti et al., 2020). The region is also especially vulnerable to climate change (Gaiser et al., 2003), and both degradation and desertification can be accelerated by gullies (Zweig et al, 2018). The Brazilian semiarid region is also characterized by shallow crystalline bedrock with scarce groundwater, which forces its population to rely almost*

*exclusively in superficial reservoirs for water supply (Coelho et al., 2017). Therefore, gullies are a two-way threat, first by depleting the already scarce groundwater and second by increasing sediment connectivity, causing supply-reservoirs siltation due to reduction of storage capacity and of water quality (Verstraeten et al., 2006).*

- The last paragraph (from L63 onwards) of the introduction was rewritten as below, in order to include a clear summary of objectives and methods.

*It is, therefore, an important milestone to understand how gully erosion starts and develops (Poesen, 2018). The objective of this work is to propose a physically-based model that simulates growing dynamics and sediment production in small permanent gullies in a hillslope scale. In order to achieve this, we tested two models – those by Foster and Lane (1983) and by Sidorchuk (1999) – and two adapted models. One modification was the insertion of a term in the Foster and Lane model, whereas the other was the coupling of both models. To validate the models, we measured the evolution of three small permanent gullies in the State of Ceará. The gullies geometry was assessed using UAV (Unmanned Aerial Vehicle).*

*We define small permanent gullies as those, which result from active erosive processes that form channels by concentrated flow and do not interact with groundwater. Normally, these gullies could be remediated by regular tillage, but in abandoned or unclaimed land, they usually remain evolving for long periods. Although the land where these erosive structures develop usually becomes useless for economic activities, the development of such gullies threatens the ecosystem and the community water supply.*

**3. Coupled Model (FL-SM):** The authors describe the governing equations of the FL model and SM model separately. It is not very clear how these equations are coupled and what platform the authors have used to run their simulation. The flow-chart showed in supplementary Fig. S3 should be moved to the main text and the modifications performed by the authors should be shown more clearly so as to ease the understanding of readers. Without that, it looks like the authors did not perform substantial modifications of the code. The evaluation of the proposed coupled model (FL-SM), along with the FL and SM models shown in Fig 6, doesn't tell the quantitative performance. I suggest the authors add the $R^2$ value for each model and verify whether a statistically significant result is gained (p-value test) or not. The same can be done for Fig 8 and Fig 11. It is also not clear why the authors have shown the rainfall comparison plots within the modelling section. Further, the explanation of Fig 5, and its relevancy to the corresponding section (modelling) is not understandable.

We agree that the present structure can be improved to increase the understanding of the processing algorithm therefore, we will change the subsection names and organization, as follows:

2.5 Gully modelling

2.5.1 Foster and Lane Model (FLM)

2.5.2 Sidorchuk Model (SM)

2.5.3 Adapted Foster and Lane Model (FLM-λ)

2.5.4 Coupled Model – Foster and Lane & Sidorchuk Model (FL-SM)

- "It is not very clear how these equations are coupled and what platform the authors have used to run their simulation"; "the modifications performed by the authors should be shown more clearly so as to ease the understanding of readers."; "Fig. S3 should be moved to the main text"

Given the new structure of this section, a detailed description of the FL-SM will be included. The flow chart showing how the coupling is performed will be included in the main text.

The coupling is governed by the threshold stablished by the cross-section area and expressed by the variable $A_c$ in the figure bellow. While the cross-section is smaller than the area threshold, the governing process is modelled using the

Foster and Lane Model. After the cross-section surpasses the threshold, the Sidorchuk Model is used. The stability of the wall is tested, and a new side slope is calculated, assuming that the cross-section area is a trapezoid.

[Figure]

- "· ". I suggest the authors add the R2 value for each model and verify whether a statistically significant result is gained (p-value test) or not. The same can be done for Fig 8 and Fig 11."

We included the R² and p-value in Figures 5, 6, 8 and 11.

[Figure]

$$\lambda = 3.4053 K_G^{-1.268}$$
$$R^2 = 0.338$$
$$\text{p-value} = 0.05$$

$$\lambda = 5.859 K_F^{0.707}$$
$$R^2 = 0.481$$
$$\text{p-value} = 0.01$$

**Figure 5** - Correlations between the ratio ($\lambda = A_o/A_m$) and (a) the Gravelius coefficient ($K_G$) and (b) the form factor ($K_F$) for 21 monitored cross-sections at MRB. Black dots refer to calibration cross-sections and white diamonds refer to validation cross-sections. The values of $R^2$ indicated in the plots are for the calibration. The validation $R^2$ were 0.10 for $K_G$ and 0.54 for $K_F$.

[Figure]

**Figure 6** - Performance of the coupled model (FL-SM), Foster and Lane Model (FLM and FLM-$\lambda$) and the Sidorchuk model (SM). P-value < 0.001 for all sets. The grey bar indicates the identified area threshold where there is a change and SM becomes consistently better than the FLM.

[Figure]

**Figure 8** - Thresholds for wall erosion: (a) based on the cross-section area; (b) based on the catchment geometry and K$_F$. In both plots the set 1 indicates the domain of bed erosion and Foster and Lane equations and set 2 the domain of wall erosion and Sidorchuk equations. p-value < 0.001.

[Figure]

**Figure 11** - Behaviour of gully growing rate as proposed by literature (Vanwalleghem et al., 2005; Poesen et al., 2011) and modelled (data from Gully 1). GV is the percentual gully volume and GT the percentual gully age.

- "It is also not clear why the authors have shown the rainfall comparison plots within the modelling section."

Figure 4 belongs to subsection *2.4 Rainfall data*. This mis-positioning was corrected.

- "Further, the explanation of Fig 5, and its relevancy to the corresponding section (modelling) is not understandable"

After implementing both FLM and SM models, we made an attempt to improve the performance of the models. The first and more simplistic was to apply an empirical factor (λ) based on the geometry of the catchment. Figure 5 presents the calibration and validation of this factor. This model is referred as FLM-λ.

With the new configuration of the subsections, this step will be more clearly described in the new point ***2.5.3 Adapted Foster and Lane Model (FLM-λ).***

**4. Model evaluations:** The authors have put a lot of effort into validating their new model. But it is very difficult to understand the model evaluator shown in Figure 10. Further explanations would suffice. The evaluation for the gully growing modelling provides satisfactory results.

We indeed made efforts to present a good validation for our results, using different indicators and graphical forms.

- In Figure 10a, we presented the evolution of the values of NSE (Nash-Sutcliff Efficiency), RMSE (Root mean squared error) and PBIAS (Percent bias) as a web plot.

- We will remove the information of the models (implementation of the FLM with different intensities - $I_{av}$, $I_{15}$, $I_{30}$ and $I_{60}$) since this is a secondary result and can be misleading. The complete plot will be moved to the Supplementary file.
- The two plots will be substituted by the plot below, which we hope to be clearer.

[Figure]

| | FLM | SM | FLM-λ | FL-SM |
|---|---|---|---|---|
| NSE | 0.56 | 0.72 | 0.69 | 0.85 |
| RSME | 0.65 | 0.44 | 0.66 | 0.40 |
| PBIAS | 18.6 | 8.5 | 11.5 | 12.6 |

Figure 10 Model evaluators NSE, RMSE and PBIAS. The web graph shows the performance of all tested models – values of PBIAS in percentage.

- Once the correlation analysis was brought to the spotlight, we suggest the inclusion of the Taylor diagram (Taylor, 2001 – "*Summarizing multiple aspects of model performance in a single diagram*")

[Figure]

**Figure 11** – Taylor diagram for the model performance. The azimuthal distance gives the correlation (R - Pearson), the distance to the origin is proportional to the standard deviation of the model values and the distance to the Reference (measured data) is proportional to the RMSE.

**4.1**: It would be better if the authors explain what methods did Poesen et al. (2011) employ.

The subsection 3.3 stating in line 277 will be updated to further explain **Figure 11**, as follows:

***3.x Gully growing modelling***

*Gully growth is commonly described as being a fast process in the first years that, then, progressively slows down its enlargement. In our model, the mechanism that produces this dynamic is event piling. It could be observed that, after a particularly intense event, the channel is sufficiently wide, whereas less intense events produce only a shallow flow with*

*low shear stress and, therefore, no erosion. Only when a more intense event then the last erosive one happens, there is further erosion. Therefore, the model also mimics the growing dynamic of gullies and its latency periods between extreme events, as reported in literature (Vanwalleghem et al., 2005; Poesen et al., 2011; Poesen, 2018) and illustrated in Fig. 11. Vanwalleghem et al. (2005), using several datasets from previous studies found a strong correlation between GT (the percentual of the gully age over the total) and GV (the percentual of the gully volume over the total), given by a function as expressed in Eq. 11. The parameters α and β were calibrated by Vanwalleghem et al. (2005) as 96.5 and -0.068 with coefficient of determination ($R^2$) equal to 0.99.*

$$GV = \alpha[1 - \exp(\beta \ GT)] \tag{11}$$

[Figure]

**Figure 11** - *Behaviour of gully growing rate as proposed by literature (Vanwalleghem et al., 2005; Poesen et al., 2011) and modelled (data from Gully 1). GV is the percentual gully volume and GT the percentual gully age.*

*The parameters α and β obtained in our study differ from the values in the literature. While α difference is due to a numerical formulation ($GV_{total}$ is equal to the measured volume), the paramentre β brings us some insights. Its value for our data set is three times bigger that calibrated by Vanwalleghem, which indicates a fast initial growth, caused by the intensive rainfall regime of the region, with convective intense events and high erosivity (Medeiros, de Araújo, 2014), a different condition from Belgium and Russia, where most studies that lead to Vanwalleghem's equation where carried on.*

**5. Discussion part:** A long discussion is provided explaining the limitations of the model and data availability, i.e., topographic, soil and rainfall data. The titles of the sub-sections could be rethought as 4.1. model limitations and 4.2 data limitations.

We thank for the suggestion of renaming the subsections. It now follows the order:
  4.1 Model limitations
    4.1.1 Foster and Lane Model
    4.1.2 Sidorchuk Model
    4.1.3 Adapted models
  4.2 Data limitations
    4.2.1 Topographic data
    4.2.2 Soil data
    4.2.3 Precipitation data

**6. English language usage:** Though the authors have put a lot of effort into the technical aspects of this manuscript, the overall writing could be improved much. I suggest the authors have a native speaker to check and rewrite the manuscript.

**Technical corrections:**
  1. Most of the figures are referred to at places after the figure is presented. The flow of the overall reading is not smooth as many figures are misplaced.

2. The authors are requested to first site the figure in the manuscript and then put it somewhere near. Also, figures are pasted in different sections, which is not relevant.

We are aware of this issue. The manuscript was written in LaTeX using the template provided by Copernicus. Although the figures and tables were positioned after citation and in their respective sections, the template modified the position in order to maximize page usage and was nonresponsive to commands such as (latex command: *[!]* ) to keep them in the desired position. We will try and modify this in the next version after the discussion is closed. In addition, we believe that during the final Editing by HESS this problem should be fixed.

3. Overall, I would suggest the authors perform professional proofreading and grammar corrections.
   We agree and it was done as requested.

4. The supplementary files just contain some figures without any description regarding how to use them whatsoever.
   We tried to organize the supplementary files in a better way, according to the respective comments associated with each figure.

5. In my opinion, this manuscript does not have major technical flaws despite the weak overall structure and complex writing style.
   We thank the Reviewer for his/her comments. We revisited the structure and writing in order to provide a better and clearer presentation of our work.

---

## Author Comment (AC2) · 16 Mar 2020

**Answer to comments of Anonymous Referee #2**

We would like to truthfully thank the Referee for his/her work, efforts and interesting observations. They helped making our work clearer.

The mains points highlighted by the Referee **Writing**, **citations**, **missing information** and **Content** . Below, we comment the suggestions point-by-point.

**Physically-based model for gully simulation: application to the Brazilian Semiarid Region**
**Dear Editor I have went through the above article. I think there are the following comments before making an acceptance decision. At present there are moderate to major comments. My decision is Major correction. Best**

We thank the reviewer for his/her comments.

**Manuscript need to check by a native for removing some language errors.**

Although much effort has been devoted to the paper writing, we acknowledge the persisting problem. We will hire a professional proof-reader for this task, after the required modifications are finished.

**Abstract I cant see any quantitative results in abstract. This part need to write again.**

The abstract was completely rewritten and the results were stated more clearly, please refer to lines 10-12.
In the box bellow we present the proposed new abstract:

Gullies lead to land degradation and desertification, an increasing environmental and societal threat especially in arid and semiarid regions, despite of which there is a lack of research initiatives in this regard. As an effort to better understand soil loss in those systems, we studied small permanent gullies, a recurrent problem in the Brazilian North-eastern semiarid region. The increase of sediment connectivity and reduction of soil moisture, among other deleterious consequences, endangers this desertification-prone region and reduces its capacity to support life and economic activities. Hereafter, we propose a model to simulate gully-erosion dynamics, derived from the previous physically-based models by Foster and Lane and by Sidorchuk. The models were adapted so as to simulate long-term erosion. A threshold area shows the scale dependency of gully erosion internal processes (bed scouring and wall erosion). To validate the model, we used three gullies ageing over six decades in an agricultural basin in the State of Ceará. The geometry of the channels was assessed using UAV (Unmanned Aerial Vehicle) and Structure-from-Motion technique. Laboratory analyses to obtain soil properties were performed. Local and regional rainfall data were gauged to obtain sub daily rainfall intensities. The threshold value (cross-section area of 2 $m^2$) characterise when erosion in the walls due to loss of stability becomes more significant than the detachment of sediments in the wet perimeter. The 30-minute intensity can be used when no complete hydrographs from the rainfalls are available. Our model can satisfactorily simulate the gully-channel cross-section area growth over time, yielding Nash efficiency of 0.85 and $R^2$ of 0.94.

**1 Introduction Lines 16-17: this sentence needs to reference.**

These sentences were rewritten and rearranged as follows:

*The impact of water-driven soil erosion, on economy and food supply alone, represents an annual loss of US$ 8 to 40 billion (Pimentel et al., 1995); a reduction in food production of 33.7 million tonnes and an increase in water consumption*

*by 48 km³ (Sartori et al., 2019). These effects are felt more severely in countries like Brazil, China and India; and in low-income households worldwide (Nkonya et al., 2016)*

**There are too many citation in this part. Please decrease them and try to just use from newest citations. Please clearly specify innovative of the current study**

A complete checking of references was conducted. Redundant citations were removed. We identified 18 non-essential references (present only to state a time-line of each specific topic).

**2.1 Study area Please add geographical coordinate of the study area.**

The coordinates will  be included:

| Area | Latitude | Longitude |
|------|----------|-----------|
| Gully 1 | 04°58'54.32"S | 39°29'36.41"W |
| Gully 2 | 04°59'53.12"S | 39°29'49.38"W |
| Gully 3 | 05° 00'02.37"S | 39°29'59.42"W |

**Lines 83-84: please add a reference. The same lines 85-86.**

The paragraph was rewritten, as shown below. The information contained in lines 81-86 can be found in Gaiser et al. (2003).

*The study area is located in the Madalena Representative Basin (MRB, 75 km2, state of Ceará, north-eastern Brazil; see Figure 1), inserted in the Caatinga biome, a dry environment with a semiarid hot BSh climate, according to the Köppen classification. The annual precipitation averages 600 mm, concentrated between January and June (Figure 2); and the potential evapotranspiration totals 2,500 mm.yr⁻¹. Geologically, the basin is located on top of the crystalline bedrock with shallow soils and limited water storage capacity. The rivers are intermittent and runoff is low, typically ranging from 40 to 60 mm.yr⁻¹ (Gaiser et al., 2003). The basin is located within a land-reform settlement with 20 inhabitants per km2, whose main economic activities are agriculture (especially Zea mays), livestock and fishing (Coelho et al., 2017; Zhang et al., 2018).*

**Figure 1. I cant see coordinate system on maps.**

We believe the referee's observation concerns Figure 1. In this Figure the Datum and coordinates are available. The sentence "Projection: UTM24S" removed, once the map is no longer in UTM coordinates (as in a previous version; we apologize for this). We hope this modification solves the problem and wait for further comments.

**2.3 Soil data How many sample do you use in this study?**

We have collected three samples at each depth (10, 30 and 50 centimetres) in both areas (S1 and S2 – line 118-119), totalling 18 samples. This information will be included in the text (lines 112-114).

**Figure 4. Correlations are so low. Why? What is its reason?**

Due to lack of better data, we used correlation curves to assess rainfall intensity based on the total daily rainfall. Rainfall processes, especially in the Brazilian semiarid regions (where most rainfalls are convective) are rather unpredictable and nonlinear processes.

**2.5.1 The Foster and Lane Model (FLM) and 2.5.2 The Sidorchuk Model (SM): These are routine and readers could find them in literatures.**

The two models are indeed well known to many readers. Nevertheless, we find it important to have a brief presentation of both models, specially to stress their differences, strengths and weaknesses. This, in our point of view, will help the readers.

However, we understand the concerns of the Referee and, therefore, both subsections will be reformulated to present frontally the models' strengths and weaknesses, reducing the space used to present them.

**You have to present your combined model very carefully. This is so important. Please edit this part.**

We thank the Reviewer very much for the suggestion. The section will be edited and clarified. We have chosen to rename and rearrange the sections in order to give it a more concise and comprehensive presentation. The new items are:

*2.5 Gully modelling*

*2.5.1 Foster and Lane Model (FLM)*

*2.5.2 Sidorchuk Model (SM)*

*2.5.3 Adapted Foster and Lane Model (FLM-λ)*

*2.5.4 Coupled Model – Foster and Lane & Sidorchuk Model (FL-SM)*

A detailed description of the proposed model was added under the item 2.5.4, including a flowchart (modified from the current Supplementary material).

**2.6 Model fitness evaluators Please add equations of the used evaluation methods.**

The equations were included as required.

**Results This part is written very carefully.**

We thank your observation and will bring the rest of the paper to these standards.

**Discussion I think it is important to add some discussions and comparisons by previous works**

We thank the suggestion of discussion and believe it will improve the debate. Firstly, we decided to rearrange the discussion section, organizing its subsections as follows:

4.1 Model limitations

    4.1.1 Foster and Lane Model

    4.1.2 Sidorchuk Model

    4.1.3 Adapted models

4.2 Data limitations

    4.2.1 Topographic data

    4.2.2 Soil data

    4.2.3 Precipitation data

In the item **4.1.3 Adapted models** we will include a paragraph comparing the quality of our model with others, as follows:

*Comparatively with other models, either physical or empirical (Haisine and Rose, 1992; Woodward, 1999; Wells et al., 2013; Dabney et al. 2015, etc), our proposed model (FL-SM) requires similar or less amount of data, little calibration (one parameter – the threshold) and is more versatile. Most models fail to account for multiple rainfall events (Foster and Lane; Woodward, 1999; Nachtergaele et al., 2001 and 2002; Torri and Boselli, 2003) and to consider multiple sources of sediment (Forter and Lane, 1983; Hairsine and Rose, 1992; Dabney et al. 2015). The FL-SM model ($R^2 = 0.94$) presented a better performance index than empirical models (e.g. $R^2$: 0.55 and 0.12 for Woodward (1999) and Wells (2013) respectively) and physical models (e.g. $R^2$: 0.87 and 0.84 for Foster and Lane (1983) and Sidorchuk (1999) respectively).*

*This enhancing in the performance can be accounted for the more detailed modelling, considering wall failure and non-rectangular cross-section.*

**5 Conclusions This part is so general. Please add some suggestion on this new model and etc…**

- The conclusion was modified to include suggestions of following works, such as:
  - Stochastic modelling to account for lack of sub-daily rainfall data and to SSY (Sediment Yield) from the catchment into the gully.
  - Inclusion of other sediment sources such as headcut retreat and flow jets.

---

## Author Response (AR1)

Ernst-Reuter-Platz 1
Gebäude BH-N, Raum 802
10587 Berlin

April 28, 2020

To the Editor Roberto Greco,

On behalf of all authors, I would like to thank the Editor and the Referees for taking their valuable time to make our work better. We fully agree with your recommendations.

We performed a complete review of the text, including change of order and structure of paragraphs and subsections. We believe that now the text is much clearer and less ambiguous. We found, after the review, that many paragraphs had disjoint information and, sometimes, the information was rather scattered between two paragraphs. We hope to have fixed this flaw.

A native speaker and professional proofreader made full revision of the text regarding grammar, structure and style, to make the language clearer. Besides, the supplement material was reformulated with the addition of more information.

We hope that this new version meets the requirements and expectations of the Hydrology and Earth System Science Journal.

Sincerely,

Pedro Alencar

[revised manuscript text omitted]

---

## Referee Report (RR1)

General comments:

The authors have performed substantial revisions based on the reviewer's comments. I thank the authors for their detailed answers to my and the other reviewer's comments and suggestions. I was concerned about the paper structure, and statistical analysis of the model performance. The revised paper and point-by-point responses are satisfying. Following my questions, the authors took the effort to change and extent parts of the text. I can suggest accepting the paper also because the authors elaborated in detail on the questions of the other reviewers, which significantly has improved the paper. My specific major suggestions and the reasons I found them okay are as follows,

Specific comments:

1. Abstract:
   As I suggested before, the abstract is rewritten in a better way clearly stating the outcomes of the proposed models.
2. Introduction:
   The introduction part is also rewritten (structurally) and two new paragraphs were added to explain the study area and the last paragraph explaining the objectives and methods clearly.
3. Coupled Model (FL-SM):
   I like the way the authors revisited the presentation of the models as in the new manuscript in section 2.5. The responses to my suggestions are indeed clear. A flow chart of the modeling approach is shown and R2 (evaluators) are added to all the plots comparing the performance of different models. Thanks.
4. Model evaluations:
   In response to my previous comment model evaluations were done using NSE (Nash-Sutcliff Efficiency), RMSE (Root mean squared error), and PBIAS (Percent bias) as a web plot. These and the new Taylor diagrams are visually appealing and suffice the reader for understanding.
5. Discussion part:
   In response to my previous comment, the discussion part is restructured as a model and data limitations. The limitations of all three models and three data sets used are pointed out clearly. It is now clear.
6. English language usage:
   Thanks. The new manuscript has been proofread and I don't have any major disagreement with the writing this time.

Technical corrections:

1. All the technical corrections were amended.

As I suggested before, this manuscript does not have major technical flaws despite the weak overall structure and complex writing style in the first submission. Since the authors revised it well based on the reviewer's comments, I suggest acceptance.

Congratulations!

---

## Author Response (AR2)

Ernst-Reuter-Platz 1
Geba¨ude BH-N, Raum 802
10587 Berlin

July 21, 2020

To the Editor Roberto Greco and the referees, we hope this letter finds you well and healthy,

On behalf of all authors, I would like to thank the Editor and the Referees for the dedication and work you have dispensed.

We are glad we met the requirements for the Hydrology and Earth System Sciences and excited to have our work published. The referees' comments were essential for the final state of our paper. For this, we are thankful.

Finally, we would like to thank you for the valuable work of editing and revising works, a task sometimes foreseen, although decisive in producing science.

We look forward for future interactions

Sincerely,

Pedro Alencar